# Innovative Therapeutic Approaches for the Treatment of the Ocular Morbidities in Patients with EEC Syndrome

**DOI:** 10.3390/cells12030495

**Published:** 2023-02-02

**Authors:** Vanessa Barbaro, Filippo Bonelli, Stefano Ferrari, Giulia La Vella, Enzo Di Iorio

**Affiliations:** 1Fondazione Banca degli Occhi del Veneto Onlus, Via Paccagnella, 11, 30174 Venice, Italy; 2Clinical Genetics Unit, University Hospital of Padua, 35128 Padua, Italy; 3Department of Molecular Medicine, University of Padua, 35128 Padua, Italy

**Keywords:** epithelial stem cells, p63, cornea, EEC syndrome, siRNA, induced pluripotent stem cells (IPSCs)

## Abstract

Ectrodactyly-Ectodermal dysplasia-Clefting (EEC) syndrome is caused by heterozygous missense point mutations in the *p63* gene, an important transcription factor during embryogenesis and for stem cell differentiation in stratified epithelia. Most of the cases are sporadic, related to de novo mutations arising during early-stage development. Familial cases show an autosomic dominant inheritance. The major cause of visual morbidity is limbal stem cell failure, which develops in the second to third decade of life. Patients often show ocular surface alterations, such as recurrent blepharitis and conjunctivitis, superficial microlesions of the cornea, and spontaneous corneal perforation and ulceration, leading to progressive corneal clouding and eventually visual loss. No definitive cures are currently available, and treatments to alleviate symptoms are only palliative. In this review, we will discuss the proposed therapeutic strategies that have been tested or are under development for the management of the ocular defects in patients affected by EEC syndrome: (i) gene therapy-based approaches by means of Allele-Specific (AS) siRNAs to correct the p63 mutations; (ii) cell therapy-based approaches to replenish the pool of limbal stem cells; and (iii) drug therapy to correct/bypass the genetic defect. However, as the number of patients with EEC syndrome is too limited, further studies are still necessary to prove the effectiveness (and safety) of these innovative therapeutic approaches to counteract the premature differentiation of limbal stem cells.

## 1. Introduction

Ectrodactyly-Ectodermal dysplasia-cleft (EEC) syndrome (MIM#604292) is a rare genetic disease caused by heterozygous, missense mutations in the TP63 gene [1]. According to www.orpha.net (last accessed on 30 November 2022), the estimated prevalence is 1:900,000, and both sporadic and familial cases are described in the literature. EEC belongs to a collection of syndromes caused by p63 mutations, including Limb mammary syndrome (LMS), acro-dermato-ungual-lacrimal-tooth syndrome (ADULT), ankyloblepharon-ectodermal defects-cleft lip/palate syndrome (AEC), and Rapp-Hodgkin syndrome (RHS), all presenting with similar/overlapping phenotypes [2]. Forty different mutations have been characterized to date, with five arginine codons being mutational hotspots (R204, 227, 279, 280, 304) [3]. Among all, the most commonly reported aminoacidic alterations are R204W/Q, R279 C/H/Q, R280C/H/S, and R304W/Q [4].

The p63 protein is a member of the p53 transcription factor family, together with p73. It is involved in several cellular functions, such as differentiation, stemness, death, migration, metastasis and senescence [5]. It works as a tetramer, hence it is supposed that in individuals affected by ectodermal dysplasia around 50% of the polymers cannot fulfill their biological function. In particular, mutations associated with EEC syndrome target the protein’s DNA binding domain, and can occur at various stages of embryonal development [6]. Dysfunctions in p63 contribute to improper epidermal development and differentiation, affecting epidermal lineage commitment, basement membrane deposition and keratinocyte adhesion [7]. In the literature, a plethora of different phenotypes have been reported; however, the most common manifestations are split-hand-split-foot phenotype, defects in tissues of ectodermal origin (skin, hair and teeth), and cleft lip and palate [8,9] (Figure 1).

In adult tissues, p63 is highly expressed in the basal layer of the epidermal tissues, where the stem and transient amplifying pool of cells are normally located. Indeed, it has been shown that animal models of p63 that are double knock-out are non-viable, as they lack the ability to generate a proper epidermis [10]. As the corneal epithelium derives from the ectoderm and shares its anatomical structure with the epidermis, EEC patients suffer from visual morbidities (Figure 2). The lower yield of p63, in particular of its ΔNp63α isoform, leads to a progressive and inexorable limbal stem cell deficiency (LSCD) [1,4]. LSCD generally manifests in the second to third decade of life, resulting in recurrent corneal ulceration, neovascularization, inflammation and spontaneous corneal perforation [1,11]. As a consequence, a dense, vascularized corneal pannus invades the cornea until opacification leads to irreversible blindness [1,12,13,14,15]. Another hallmark of the pathology is the involvement of ocular adnexa, with partial or complete Meibomian gland agenesis and defects in the lacrimal ducts which are commonly reported in EEC patients [15,16,17]. An insufficient hydration triggers the exfoliation of the corneal epithelium, thus further stressing the replicative activity of the limbus, eventually resulting in its early exhaustion.

In this review, we aim at dissecting the current state of the art of the proposed therapeutical approaches for the management of the ocular manifestations in patients with EEC syndrome, shedding some light on the new technologies that are under development, including stem cell-, gene therapy- and drug therapy-based strategies.

## 2. Corneal Transplantation Does Not Lead to Any Benefit

The first attempts to manage the progression of ocular disease in EEC patients mostly relied on penetrating keratoplasty. Indeed, before it became evident that LSCD is the driving mechanism leading to vision failure in EEC subjects, surgeons considered corneal graft as the standard of care for complications including corneal perforations, melting or opacification. As it is now clear that penetrating keratoplasty is inappropriate for the management of ocular complications in patients with EEC syndrome, and that such an approach can instead have devastating results, this section will review a few cases reported in the literature, including their respective post-surgery outcome.

To the best of our knowledge, the first case of corneal transplantation in a patient with EEC syndrome was reported in 1972, on a girl aged 13 with marked corneal opacity [18]. Following the intervention, however, clouding and neovascularization invaded the graft.

In 1974 another case was described, a young girl aged 5 who underwent corneal transplant as a result of a sterile ulcer. The outcome was similar, with a graft that failed to re-epithelialize and then rapidly neovascularized and opacified [19].

Another paper reported the case of two subsequent penetrating keratoplasties performed on a patient with corneal melting. In both cases the graft failed, with recurrence in corneal melting [20].

In 1990, two cases—a 45-year-old woman and her 23-year-old son—of corneal transplantation on EEC patients due to sterile corneal meltings were described by Mader and Stulting [21]. To date, this is the longest follow-up (22 months) described in the literature with low to mild signs of clouding and neovascularization. Curiously, these two patients were reported to have functional meibomian glands, contrary to what is commonly observed in EEC patients.

An interesting case was described in 2003, reporting the case of a 25-year-old woman that underwent her first penetrating keratoplasty in the right eye at the age of 18 because of progressing corneal scarring [22]. Follow-up analyses of the corneal button showed a thin, irregular epithelium, the absence of Bowman’s membrane, scarring and fibrosis at the midstromal level, inflammatory infiltrate, and neovascularization. The left eye, presenting with vascular pannus and superficial scarring, underwent a superficial keratectomy to remove the fibrous layer and vessels first, and four penetrating keratoplasties afterwards. All of them failed, except for the last surgery that made use of a keratolimbal allograft. At the last follow-up, one year after the transplant, the corneal button was still transparent with traces of mild neovascularization. Similarly, a second penetrating keratoplasty with keratolimbal allograft was performed in her right eye, with a similar outcome to that observed in the left eye. In the same paper, a further case of a 56-year-old woman is described. She received a bilateral corneal transplantation with keratolimbal allografts for the management of sterile corneal perforations. In both eyes, corneal buttons started to show signs of stromal scarring, loss of lamellar architecture, discontinuous Bowman’s membrane and hypercellular stroma, neovascularization, and the infiltration of inflammatory cells a few months after the transplant.

In 2003 a case of a woman aged 28 with recurrent ocular infections was reported. She required penetrating keratoplasty to manage her clinical situation, which was probably the result of a bilateral lacrimal duct obstruction as a contributing cause [23]. However, no follow-up results were published.

Finally, we recently came across the case of a 62-year-old subject with EEC syndrome who underwent two subsequent corneal transplants due to an infection in the right eye following cataract surgery. Also in this case, both grafts failed and the corneal epithelium eventually showed signs of conjunctivalization (unpublished data). Information about the EEC patients undergoing penetrating keratoplasty is shown in Table 1.

## 3. New Approaches for the Management of the Ocular Defects in Patients with EEC Syndrome

As corneal transplantation showed not to lead to any benefit in patients with EEC syndrome, different approaches have started to be developed with the aim of correcting the genetic mutations in the p63 gene, replenish the pool of stem cells or correct/bypass the genetic defect through drugs. Such innovative strategies are described below and are briefly summarized in Figure 3.

### 3.1. Allogeneic Limbal Stem Cells

The adverse events related to long-term systemic immunosuppression and the risk of rejection associated with techniques such as Keratolimbal Allograft (KLAL) [22] and Conjunctival Limbal Allograft (CLAL) [24] transplantation, along with the relatively large amount of donor tissue required, have been the rationale for the use of cell-based therapies to treat LSCD. Ex vivo Cultured Limbal Epithelial Transplantation (CLET) involves the in vitro expansion and differentiation of limbal epithelial stem cells to form an epithelial sheet which is subsequently applied to the cornea with or without an underlying substrate, for example, amniotic membrane or fibrin glue [25]. CLET has the benefit of avoiding large tissue biopsy and transplantation as in KLAL and CLAL, while transplanting only those limbal stem cells/progenitor cells that are capable of in vitro expansion, thereby improving the chances for successful epithelial reconstruction.

Owing to the presence of *p63* mutations, in patients with EEC syndrome and ectodermal dysplasia, CLET has been performed by using unaffected allogeneic donor cells and with the recipient undergoing immune suppression to prevent rejection of the allogeneic graft. Examples of allogeneic CLET in patients with EEC syndrome or, more generally, with ectodermal dysplasia, are limited to just a few cases that are described below.

In 2008, Shortt and colleagues reported the outcome of ex vivo cultured limbal epithelial stem cell transplantations in 10 eyes of 10 patients with LSCD, including *n* = 1 with ectodermal dysplasia (but no indications of whether this was an EEC syndrome were given—see details in Table 2). The patient, a 32-year-old female, received a second allograft 1 month after the initial one because of failure of the first graft. Six months later, post-operative visual acuity did not improve (perception of light) and persistent epithelial defects were still observed. No impression cytology or confocal microscopy-based analyses were performed post-operatively, and the overall clinical outcome was defined as “failure” [26].

In a similar study, Daya and colleagues investigated the fate of allogeneic limbal stem cells following transplantation in 10 eyes of 10 patients, with *n* = 2 having LSCD arising from EEC syndrome and ectodermal dysplasia. The first patient, a 3-year-old female with ectodermal dysplasia, had previous amniotic membrane transplantation and received allogeneic tissue from a living relative. Principal indications for surgery were persistent epithelial defects and poor vision. At the last follow-up (27 months), she had improved visual acuity (to 20/160) after surgery, despite amblyopia. The clinical outcome was defined as successful, even if her preoperative vision acuity could not be established before surgery due to age and photophobia. The second patient, a female aged 31 years, with EEC syndrome, had many previous surgical procedures, including Deep Anterior Lamellar Keratoplasty, KLAL, amniotic membrane transplantation and living related CLAL—all unsuccessful. Principal indications for surgery were persistent epithelial defects. Preoperative visual acuity was 4/200. After an initial success with a stable ocular surface until 26 months, she developed a persistent central epithelial defect 1 month before the most recent follow-up. Failing to demonstrate an overall improvement in parameters of LSCD, she was classified as a failure. In both patients, a DNA analysis of impression cytology specimens between 1 and 7 months postoperatively revealed the presence of only host DNA, thus raising questions about the origin of the host corneal epithelium [27]. Information about the patients with EEC syndrome and ectodermal dysplasia undergoing allo-CLET is shown in Table 2.

In summary, the bilateral LSCD in EEC syndrome precludes an autologous source of healthy limbal stem cells (e.g., Holoclar). The very few cases reported above indicate that while vision rehabilitation is possible in the short term, in the longer term allo-CLET seems to fail as a treatment for patients with EEC syndrome or, more generally, ectodermal dysplasia.

The problem of immunosuppression remains similar to KLAL, although it was speculated that there might be a reduced risk of allograft rejection when using ex vivo cultivated cells, explained by the absence of antigen-presenting Langerhans cells. The lack of evidence for allogeneic limbal stem cell survival beyond a relatively short period of time [27] prompts important questions about the type and length of immunosuppression. However, different from life-saving organ transplantation protocols, in allo-CLET the benefits of current immunosuppression regimens will need to be balanced against the risks of developing side effects in the longer term.

Similarly to organ transplantation, Human Leukocyte Antigen (HLA) matching has also been sought as a strategy to reduce rejection. The Cincinnati Protocol describes the pre-operative screening and donor selection algorithms before considering KLAL or living related CLAL as a way to minimize the antigenic burden and select the best available donor match [28]. This led Behaegel and colleagues to evaluate the outcomes of an allogeneic HLA-matched allo-CLET for the treatment of aniridia-associated keratopathy in 6 patients with aniridia, a rare genetic disorder due to mutations in the Pax6 gene [29]. However, out of 6 eyes, 4 were graded as failure and the remaining one was partially successful. The authors therefore concluded that HLA matching was insufficient to prevent a high incidence of post-operative persistent epithelial defects and ultimately CLET failure. There is no evidence to believe that the same would not occur in patients with EEC syndrome and therefore alternative immunosuppression protocols or stem cell-based approaches (see Section 3.2 and Section 3.3 below) will have to be developed.

### 3.2. Oral Mucosal Epithelial Stem Cells

In recent years, patients with chemical and thermal burns, Stevens-Johnson syndrome, mucous membrane pemphigoid and idiopathic ocular surface disorders have been treated using a technique known as Cultivated Oral Mucosal Epithelial Transplantation (COMET) [30,31]. The technique requires the enzymatic treatment of oral mucosal epithelial cells that are obtained from a buccal mucosal biopsy from the patient. These cells are then cultured and stratified onto an amniotic membrane, which is thereafter transplanted on the denuded corneal surface.

Oral mucosal epithelial stem cells have never been used for the treatment of LSCD in patients with EEC syndrome or ectodermal dysplasia due to the expression of the mutated *p63* gene.

However, Barbaro and colleagues recently described the case of a young female patient, aged 18 years, with EEC syndrome, who was homozygous for a novel and de novo R311K missense mutation in the *p63* gene [11]. A detailed analysis highlighted the presence of a somatic mosaicism, with approximately 80% of cells being homozygous and 20% heterozygous. A likely hypothesis is that the somatic mosaicism combined with (A) a milder severity of the mutation when heterozygous and (B) a heterozygosity of at least 20% of cells, contributed to the survival of the patient. Oral mucosa epithelial stem cells (OMESCs) carrying the R311K mutation were expanded in vitro and heterozygous holoclones selected following clonal analysis. In vitro, such cells generated an epithelium, which was well organized and stratified into 4 to 5 cell layers, resembling the features of healthy tissues. In sharp contrast, tissues generated from OMESCs carrying more severe p63 mutations, such as R279H and R304Q, showed defects in both stratification and differentiation, with a lack of proper tissue polarity. Such findings strongly support the rationale for the development of grafts obtained by culturing autologous heterozygous R311K-p63 OMESCs as an effective therapy for reconstructing the ocular surface, thus bypassing any gene therapy approach.

However, while promising, such a therapeutic approach will be limited to the unique patient with EEC syndrome identified by the authors and therefore other strategies (cell-, drug- or gene-therapy based) will have to be identified for clinical applications involving patients with other p63 mutations.

### 3.3. Alternatives to Allogeneic Primary Limbal Stem Cells 

Limitations of allo-LSCT have motivated the investigation of alternatives to primary Limbal Stem Cells (LSCs). Amongst these, induced pluripotent stem cells, ABCB5+ cells and mesenchymal stem cells are at the forefront of research and will be discussed here in more detail.

#### 3.3.1. Induced Pluripotent Stem Cells

The unmet clinical need for immune-compatible LSCs is challenging researchers to consider novel cell sources. The differentiation of LSCs from human pluripotent stem cells (hPSCs), including both human embryonic stem cells (hESCs) and induced pluripotent stem cells (iPSCs), may represent a promising therapeutic option, especially for patients suffering from bilateral LSCD, such as the EEC syndrome. For therapeutic use, the sourcing of autologous cells from EEC syndrome patients for reprogramming is likely not an option because of the underlying genetic defect, but the use of allogeneic and standardized therapeutic-grade iPSC could be an area for investigation, especially if the immunogenicity problems of allogeneic iPSC or hESC-derived cells could be overcome with a combination of adequate HLA-matching and short-term immunosuppression.

Very recently, the first patient was treated with iPSC-differentiated corneal epithelial cells [32], thus suggesting that the technology is ready for clinical applications. In addition, the possibility of generating corneal cells and corneal organoids from patient-specific iPSCs and also deriving iPSC lines carrying specific corneal disease mutations will allow to have in vitro models that are able to recapitulate the molecular bases of any given pathology. In the future, such tools might become crucial to (A) dissect the molecular mechanisms of a disease, (B) facilitate drug discovery/targeting, and (C) develop new cell/gene therapy-based approaches.

With such aims, Trevisan and colleagues described the generation of integration-free EEC-hiPSCs by reprogramming oral mucosa epithelial stem cells from a healthy subject and two EEC patients (with R279H and R304Q mutations) by means of a Sendai viral vector and episomal vector-based reprogramming. Both healthy and mutated hiPSCs differentiated towards the corneal epithelium with the expression of markers such as dΔaNp63alpha, ABCB5 and keratin 12, even if at lower levels compared to primary limbal epithelial stem cells [33,34,35,36]. Similarly, as described in greater detail below, Shalom-Feuerstein et al. reprogrammed fibroblasts from healthy donors and EEC patients carrying two different point mutations in the DNA binding domain of p63 into iPSC lines and managed to rescue and revert the impaired epithelial differentiation following the application of APR-246 (PRIMA-1^MET^) [37].

Whether these approaches will be clinically relevant remains to be demonstrated. However, hiPSC-LSCs may represent a prospective new source for ocular surface reconstruction in patients with bilateral LSCD, as in the EEC syndrome, but critical preclinical safety and efficacy evaluation of these cells is crucial before translation to clinical applications.

#### 3.3.2. ABCB5+ Cells and Mesenchymal Stem Cells

To overcome the issues associated with the use of allogeneic cells and immunosuppression regimens, a further strategy might rely on the use of ABCB5+ LSCs and mesenchymal stem cells (MSCs) as advanced therapy medicinal products (ATMPs).

ABCB5, a membrane-bound ATP-binding cassette transporter, subfamily 5, member 5, has been described as the first molecular surface marker for prospective LSC enrichment by antibody-based cell sorting [38]. In contrast to ΔNp63α, which does not allow any prospective cell sorting-based enrichment of LSCs given its nuclear localization, ABCB5 is a molecular surface marker. As reported previously, clinical studies using allogeneic limbal tissue transplants have so far only provided transient corneal restoration. It is thought that one of the reasons for such failures might be the presence of immunogenic limbal cell subpopulations, such as Langerhans’ cells, capable of inducing rejection responses in the recipient. The LSCs used in such trials, in fact, comprise only a small population among heterogeneous cell populations present in the limbus. The possibility to isolate, expand and purify ABCB5+ LSCs from deceased donors through antibody-based cell sorting might overcome these obstacles by precluding the transfer of potentially highly immunogenic cell subpopulations, ensuring defined composition and purity of the cell product. The use of GMP-compliant ABCB5+ cells might be a new strategy for the treatment of bilateral LSCD, as in the EEC syndrome, which requires allogeneic LSC transplantation. Importantly, their use might require reduced immunosuppression regimens, since preliminary data indicate that pure populations of ABCB5+ stem cells are minimally immunogenic. As a consequence, unwanted side effects due to systemic immunosuppression could probably be avoided [39].

A further strategy might be to rely on the use of MSCs. These were first described as a rare, non-hematopoietic stem cell population in the bone marrow, but have been subsequently found in many other tissues such as adipose tissue, umbilical cord, dental pulp, conjunctival tissue and limbal stroma. MSCs are multipotent and have the potential to differentiate into various cell types, even if only very few reports have been published on MSC differentiation into a corneal phenotype. However, what is believed to be the major therapeutic benefit of MSC transplantation is probably not the potential to differentiate into various tissues, but rather the capacity to modulate immune responses [40]. Applying this concept to allo-LSCT, the co-transplantation of allogeneic MSC and LSC could therefore be an interesting therapeutic option to alleviate allogeneic immune responses. In a recent pilot clinical trial, the safety and efficacy of allogeneic bone marrow-derived MSC was compared to allo-CLET to treat LSCD. Results showed that the bone marrow-derived MSC application was safe and as efficacious as allo-CLET, and no adverse events related to cell products were recorded [41].

For both strategies (ABCB5+ cells and MSCs), no clinical trial has been carried out in EEC patients. However, due to their potentially lower immunogenicity and immunomodulatory properties, they might constitute a novel therapeutic concept to improve LSCD in such patients.

### 3.4. Gene Therapy-Based Approaches 

Allele-specific (AS) siRNAs silencing may represent a potential therapeutic approach for autosomic dominant syndromes, and has already been evaluated for the treatment of Meesmann epithelial corneal dystrophy [42,43] and Lattice corneal dystrophy type I [44], as well as for dermatological disorders such as epidermolysis bullosa simplex [45] and epidermolitic palmoplantar keratoderma [46]. As EEC syndrome results from heterozygous dominant-negative mutations in the p63 gene, AS gene silencing through RNAi is a viable option to specifically inhibit the expression of the disease-associated allele without suppressing the expression of the corresponding wild-type (WT) copy.

Currently, a specific locked nucleic acid (LNA) modified siRNA (named siRNA *a*), designed to target the R279H-ΔNp63α allele, has been identified and shown to downregulate the R279H-ΔNp63α mRNA in oral mucosal epithelial stem cells from EEC patients by approximately 80%, while the corresponding WT was stably expressed and unaffected [47].

The downregulation was assessed by means of an allele-specific Real-Time PCR, a sensitive assay developed to detect the R279H mutation in the *p63* gene, and therefore useful as a tool to evaluate the effectiveness of therapies aimed at reducing the levels of mutated allele [48].

After siRNA treatment, compared to controls, mutated OMESCs exhibited a longer acquired life span, with a less accelerated stem cell differentiation in vitro and reduced proliferation. Furthermore, the correction of epithelial hypoplasia was observed in a model of organotypic culture, thus resulting in a full thickness stratified, well organized and differentiated epithelium characterized by basal expression of p63 and ΔNp63a while keratin 3 and 14-3-3 σ were mainly found expressed in the upper cell layers. The basal cuboidal cells were anchored to the basement membrane and expressed β3-laminin [47].

Such results support the application of mutant-specific siRNA molecules to obtain the phenotypic correction of mutant EEC epithelial cells with restoration of their functions. Since p63-defective-limbal epithelial stem cells (LESCs) show a reduced ability to repopulate the corneal epithelium, in young patients, who still have LESCs in the limbus, the use of eye drops containing mutant-specific siRNAs may be a practical therapeutic option to counterbalance the loss of stem cells.

In a similar study, Novelli et al. [49] selected two effective siRNAs for ΔNp63-R304W EEC mutants (T4/T11), and tested their ability to rescue ΔNp63α-WT transcriptional activity in induced pluripotent stem cells derived from skin biopsies of EEC patients. WT and EEC-iPSCs were differentiated in corneal cells in order to test whether mutated allele-specific siRNAs were able to restore p63 function and promote corneal differentiation. After siRNA treatment, the EEC-iPSCs corneal epithelial differentiation was partially rescued, thus demonstrating that a siRNA strategy could be promising in order to delay the loss of corneal function.

### 3.5. Drug-Based Therapies

While in childhood clefting and hand deformities are the main clinical features, during early adulthood the ocular problems become the predominant clinical feature of EEC syndrome [50]. EEC-related corneal pathology follows a clear clinical course, with limbal stem cell deficiency leading to severe corneal failure in the fourth to fifth decade of life. This provides a useful therapeutic window for testing new pharmacological therapies aimed at correcting or bypassing the genetic defect.

#### 3.5.1. APR246/PRIMA-1^MET^

Given the high sequence and functional homology between p53 and p63, it has been shown that the small compound APR246/PRIMA-1^MET^, capable of restoring the wild-type conformation to mutant p53 and inducing apoptosis in cancer cells, (clinical trial phase I/II, [51]), was also able to re-establish the p63 activity in primary cells of EEC patients. Two cellular models were tested, both obtained from EEC patients carrying the mutations R304W and R204W: corneal epithelial cells derived from fibroblasts reprogrammed into iPSCs [37], and skin keratinocytes [52]. While the IPSC^EEC^ failed to differentiate into corneal and limbal epithelial cells, the treatment with APR246/PRIMA-1^MET^ restored the corneal epithelial commitment both in IPSC^R304W^ and IPSC^R204W,^ with a normal p63-related signalling pathway.

In addition, it has been shown that APR246/PRIMA-1^MET^ partially restored the epidermal differentiation of adult skin keratinocytes from EEC patients carrying the R204W and R304W mutations in 2D submerged cultures and in 3D human skin equivalents, likely through the restoration of p63 target gene expression.

On the basis of the in vitro results and the ongoing phase II APR246/PRIMA-1^MET^ trial in cancer patients, the compound was topically administered to two patients affected by AEC syndrome (a p63-related ectodermal dysplasia with mutations found in the sterile α-motif or transactivation-inhibitory domain and also characterised by alopecia and skin erosion), with a significative improvement in epidermal covering, thus suggesting an effect of the compound on defective wound healing [53].

These results might pave the road for testing the drug as eye drops on the ocular surface of patients with EEC syndrome and for evaluating whether APR246/PRIMA-1^MET^ is effective in restoring the normal function of p63. However, preliminary proof-of-concept studies should be performed on in vitro and in vivo models of EEC syndrome before moving forward.

#### 3.5.2. DAPT (N-[N-(3, 5-Difluorophenacetyl)-L-Alanyl]-S-Phenylglycine T-Butyl Ester)

It is known that mutations in the *p63* gene induce a rapid exhaustion of the clonogenic and self-renewal potential of epithelial stem cells, resulting in premature cell senescence.

The γ-secretase inhibitor DAPT (N-[N-(3, 5-Difluorophenacetyl)-L-Alanyl]-S-Phenylglycine T-butyl ester) is an indirect Notch signaling inhibitor that is responsible for inducing the differentiation of adult epithelial stem cells [54,55]. It has been shown that Notch activity is downregulated in the corneal epithelium during the early phase of proliferative conditions [56]. Moreover, Notch inhibition using DAPT was found to accelerate corneal epithelial wound closure in an in vivo murine model without affecting proliferation [57]. A number of Notch inhibitors are currently used in clinical trials as cancer therapies [58,59], making them potentially useful for the clinical application to the cornea.

Barbaro et al. showed that the administration of DAPT to oral mucosa epithelial stem cells from EEC patients carrying the R279H mutation led to an enrichment in epithelial stem cells and the substantial extension of their lifespan. While untreated cells could be maintained in culture for 7 ± 2 passages, DAPT-treated cells were instead, were cultured for 18 ± 2 passages, about three times more than the normal length of the lifespan of such cultures [60]. A karyotype analysis performed on EEC-OMESCs treated with DAPT did not reveal any numerical or structural chromosomal abnormalities, while an increased telomere length was observed, thus confirming the correlation between the elongation of telomeres and increased cellular half-life. Moreover, the administration of DAPT resulted in a low mitochondrial activity that was reported to be linked to a higher stem cell potential in ex vivo experiments [61]. After DAPT was removed from cell cultures, the morphology of the cells appeared similar to that observed in end-stage cultures, thus excluding any immortalization process being induced by the treatment with DAPT. As described earlier, the quantitative allele-specific Real-time PCR assay [48] may also be used to quantify the *p63* mutational load after the administration of DAPT.

Such findings seem to indicate that the use of DAPT could slow down the senescence of epithelial stem cells from patients with EEC syndrome by extending their replicative capacity, thus suggesting a new potential pharmacological opportunity for the treatment of these patients.

## 4. Conclusions and Future Perspectives

The treatment of congenital LSCD in patients with EEC syndrome remains an unmet clinical need. Techniques such as Simple Limbal Epithelial Transplantation (SLET) [62], CLET and conjunctival-limbal autografting (CLAU) are inappropriate for the management of this condition, as patients lack a healthy autologous source of cells. Likewise, KLAL and CLAL represent a short-term solution, as the graft repopulates the epithelium without rooting in the palisades [27,63,64]. These techniques do require immunosuppression, and, in addition, the trauma of such surgeries in eyes affected by LSCD can further stimulate a rapid conjunctivalization and opacification of the cornea if there is no effective engraftment.

New strategies and approaches are therefore required, as highlighted in Figure 4.

The transplantation of a corneal epithelium generated from the patient’s own cells following genetic modification or correction does seem to be the only way forward for the treatment of the ocular morbities in the EEC syndrome. If the pathology is identified earlier in life, young patients may still have residual pools of limbal stem cells to be used as a source of autologous cells, and, if exhausted, OMESCs might be a challenging therapeutic alternative [30,31,65,66,67,68,69].

Due to the heterozygous, dominant-negative nature of the disease, an interesting strategy might be the addition of a wild-type copy of the p63 allele into the patient’s own cells. Via a simple viral infection, and subsequent analyses to evaluate the integration sites, it would be possible to provide the cells with a proper amount of p63 wild-type copies. It is conceivable that this would allow the cells to replicate for a greater number of cycles, thus increasing their lifespan and hence graft survival. A critical point would be the choice of the promoter, in order to balance the amount of transcript without exceeding the physiological levels of p63. In addition, clone selection should ideally be performed, as every cell may have a different viral vector integration profile (sites of integration and number of integrations) and therefore genotoxicity effects.

Another strategy might rely on the use of the Clustered Regularly Interspaced Short Palindromic Repeats (CRISPR)/CRISPR-Associated Protein 9 (Cas9), a novel technology which can allow precise gene editing at a single base level [70]. CRISPR/Cas9 has been effectively employed in research for gene knock-in and knock-out. Such a feature could be helpful to edit the mutated allele, removing the single nucleotide variant and replacing it with the correct nucleotide, thus allowing the production of functional transcripts and proteins. However, a major limitation of this technique might be caused by its low efficiency.

A recent study from Roux and colleagues reported that the administration of recombinant Pax6 fused to a cell penetrating peptide to limbal stem cells carrying a heterozygous mutation in Pax6 was able to rescue the lack of functionality. Heterozygous mutations in Pax6 are responsible for aniridia, a pathology characterized by congenital LSCD [71]. Due to the similar nature of LSCD in EEC syndrome and aniridia, another therapeutic approach could be the administration of eyedrops containing recombinant p63 fused to a cell penetrating peptide to restore limbal stem cell functionality. An issue of this approach is that, in absence of a stable correction, it would represent a lifelong treatment. Thus, this solution might represent an opportunity for the short-term management of ocular disease.

Further studies are still necessary in order to evaluate the effectiveness (and safety) of these innovative therapeutic approaches to counteract the premature differentiation of limbal stem cells in EEC patients. In addition, as the number of patients with EEC syndrome is too limited, national and international registries should be developed and implemented in order to have larger cohorts of subjects included in clinical studies needed to prove the real efficacy of the approaches outlined above.

## Figures and Tables

**Figure 1 cells-12-00495-f001:**
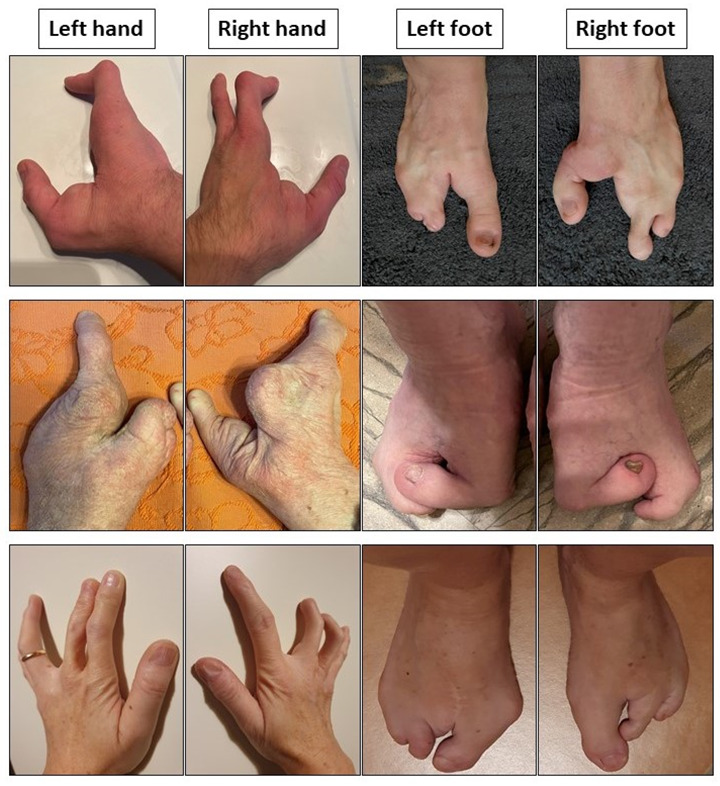
Phenotypic tracts of ectrodactily-ectodermal dysplasia clefting (EEC) syndrome. Ectrodactily and sindactily are visible in the hands and feet of three EEC patients.

**Figure 2 cells-12-00495-f002:**
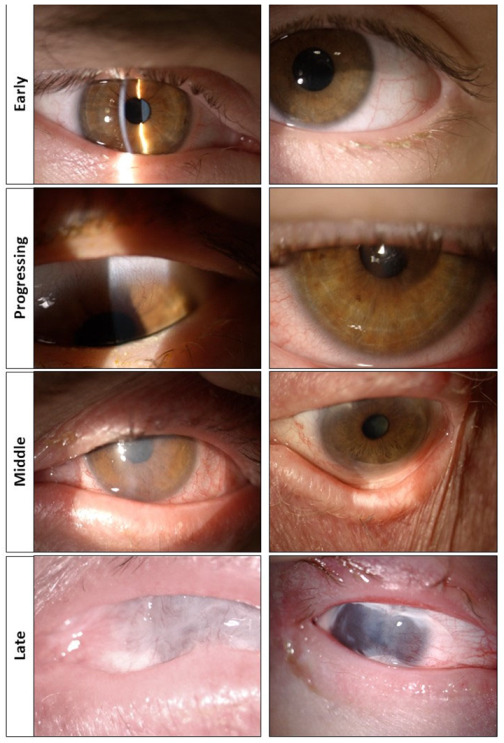
Ocular phenotype in patients with Ectrodactily-Ectodermal dysplasia Clefting (EEC) syndrome and stages of the pathology. No signs of corneal pannus or inflammatory infiltrates are present in the cornea, which still appears transparent (early stage). Traces of corneal hyperemia and neovascularizazion are starting to be visible, with vessels beginning to invade the cornea. The palisades of Vogt are almost absent (progressing stage). The corneal pannus invades the cornea with neovascularization progressing from both sides of the eye. The palisades of Vogt are absent (middle stage). Keratopathy with dense vascularized corneal pannus and symblepharon of the internal cantus are present. The palisades of Vogt are completely absent (late stage).

**Figure 3 cells-12-00495-f003:**
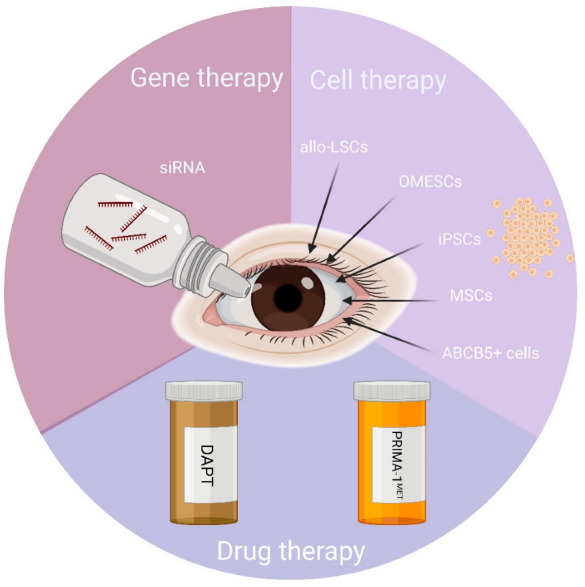
Approaches for the management of ocular defects in patients with EEC syndrome. There are a few therapeutic strategies that have been tested or are under progress for the treatment of the ocular defects in EEC syndrome: (i) gene therapy-based approaches by means of Allele-Specific (AS) siRNAs to correct the p63 mutations; (ii) cell therapy-based approaches to replenish the pool of limbal stem cells; and (iii) drug therapy to correct/bypass the genetic defect. Abbreviations: allo-LSCs: allogeneic Limbal Stem Cells; OMESCs: oral mucosa epithelial stem cells; iPSCs: Induced Pluripotent Stem Cells; MSCs: Mesenchymal Stem Cells; ABCB5: ATP Binding Cassette Subfamily B Member 5; DAPT: (N-[N-(3, 5-Difluorophenacetyl)-L-Alanyl]-S-Phenylglycine T-butyl ester); PRIMA-1MET: p53 reactivation and induction of massive apoptosis. Created with BioRender.com.

**Figure 4 cells-12-00495-f004:**
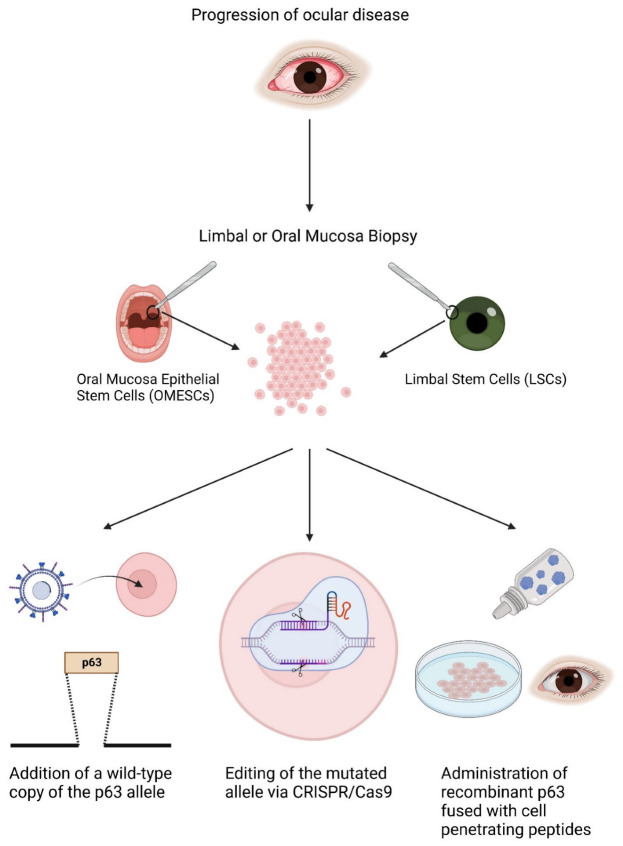
Innovative and future strategies for the treatment of the ocular defects in patients with EEC syndrome. New strategies have been suggested as a way to restore limbal stem cell functionality including (i) the addition of a wild-type copy of the p63 allele into the patient’s own cells by means of viral gene transfer agents; (ii) p63 mutated allele gene editing by means of CRISPR/Cas 9; and (iii) the administration of eyedrops containing recombinant p63 fused with a cell penetrating peptide. Abbreviations: CRISPR/Cas 9: Clustered Regularly Interspaced Short Palindromic Repeats/Associated Protein 9. Created with BioRender.com. FIGURES AND TABLES.

**Table 1 cells-12-00495-t001:** Details, follow-up time and clinical outcome of EEC syndrome patients receiving penetrating keratoplasty.

Reference	Cases	Age	Gender	Follow-Up Time	Outcome
*Corneal transplantation*					
[18]	1 patient	13	F	9 years	Graft opacification, neovascularisation, partial blindness
[19]	1 patient	5	F	1 year	Graft opacification, re-epithelialisation, neovascularisation, partial blindness
[20]	1 patient	33	F	unavailable	Corneal melting and perforation
[21]	2 patients	45 23	F M	10 months 22 months	Staphylococcal ulcerative keratitis followed by corneal perforation, secondary penetrating keratoplasty, clear vision; Mild epithelial erosion, marginal scarring and neovascularisation
[22]	2 patients	25 56	F F	14 months 6 months	Penetrating keratoplasty was repeated 5 times in left eye and 2 times in right Eye due to corneal perforations, clear vision and mild peripheral neovasularisation; Stromal scarring, loss of normal lamellar architecture, discontinuous Bowman’s membrane and hypercellualr stroma with a few chronic Inflammatory cells
[23]	1 patient	28	F	unavailable	Unavailable
unpublished	1 patient	62	M	7 years	Symblepharon/ankyloblepharon

**Table 2 cells-12-00495-t002:** Characteristics, treatments and clinical outcomes of patients with EEC syndrome/ectodermal dysplasia undergoing allo-CLET.

Reference	Age and Gender	Pre-Operative Visual Acuity	Post-Operative Visual Acuity	Dosage of Cyclosporin a	Follow-Up Time	Clinical Outcome	Further Information
[26]	32, female	PL	PL	3.5 mg/kg for 6 months	6 months	failure	
[27]	3, female	unknown	20/160, amblyopia	nil	27 months	success	DNA from host only at months 1 and 6
[27]	31, female	4/200	4/200	3 mg/kg tapered to 2 mg/kg after 2 weeks, indefinitely	27 months	failure	DNA from host only at month 7

VA: visual acuity; PL: perception of light.

## Data Availability

Not applicable.

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
