# Peer review of "Innovative Therapeutic Approaches for the Treatment of the Ocular Morbidities in Patients with EEC Syndrome"

_cells, 2023, doi:10.3390/cells12030495_

Round 1

Reviewer 1 Report

This paper investigated proposed therapeutical approaches for the management of the ocular manifestations in patients with EEC syndrome. However, their paper is not well organized, and they need further evidences to support their proposals and conclusion.

Specific major points:

1. Their abstract was too simple and many repeated sentences from Introduction, which should be revised.

2. It is a REVIEW ARTICLE. They need to revise too short Introduction session, including detailed pathogenesis, related with clinical symptoms/severity, genetic variation, and detailed genetic mutation

3. In general, most bibliography is out of date, few references from recent years. Could it be updated?

4. The authors should add several photos and presentation about their cases of EEC on page 4.

5. What is the meaning of “prompts ethical questions regarding the continuation of immunosuppression after nine months,” on page 6? What about other organ transplants?

6. Plz, add MHC related problems about allogeneic limbal stem cells.

7. The authors need to add more Gene therapy-based approaches.

8. They should explain the therapeutic limitations and drawbacks of each treatment options, including future perspectives part, in detail.

9. Plz, add summarized tables about all therapeutic modalities And also differentiation molecular methods of iPS cells.

10. There are articles published in the last year that talk about other therapies that could serve to broaden and update the discussion.

11. Future lines of therapeutic option should be expanded.

12. What is the major difference of EEC and generally known limbal cell deficient diseases, regarding on treatment and prognosis? Why?

Author Response

Response to Reviewer 1 Comments

Point 1: Their abstract was too simple and many repeated sentences from Introduction, which should be revised.

Response 1. We agree with the reviewer. We have therefore modified the Abstract and replaced the repeated sentences.

Point 2. It is a REVIEW ARTICLE. They need to revise too short Introduction session, including detailed pathogenesis, related with clinical symptoms/severity, genetic variation, and detailed genetic mutation

Response 2. We thank the reviewer for his/her comments. We have expanded the Introduction session.

Point 3. In general, most bibliography is out of date, few references from recent years. Could it be updated?

Response 3. As suggested by reviewer, the bibliography has been updated (the added references are highlighted in yellow).

Point 4. The authors should add several photos and presentation about their cases of EEC on page 4.

Response 4. We thank the reviewer for his/her comments. We added 4 additional photos of different stages of the pathology in Figure 2.

Point 5. What is the meaning of “prompts ethical questions regarding the continuation of immunosuppression after nine months,” on page 6? What about other organ transplants?

Response 5. The meaning of the sentence has been explained better and at lenght in lines 214-220 of paragraph 3.1, with specific references to organ transplants.

Point 6. Plz, add MHC related problems about allogeneic limbal stem cells.

Response 6. The results of HLA-matched allo-CLET protocols have been reported and described in lines 221-232 of paragraph 3.1.

Point 7. The authors need to add more the Gene therapy-based approaches.

Response 7. The approach to correct the genetic defect of p63 in EEC patients described in the present review (allele-specific siRNAs silencing) refers to strategies aimed to specifically inhibit the expression of the mutated allele without suppressing the expression of the corresponding wild-type (WT) copy. Currently, this represents the only proposed treatment in vitro (Peter J. Koch and Maranke I. Koster: Rare Genetic Disorders: Novel Treatment Strategies and Insights Into Human Biology; Front Genet. 2021, 12:714764. doi: 10.3389/fgene.2021.714764), though it still needs to be verified in terms of delivery, long-term efficacy and safety.

Point 8. They should explain the therapeutic limitations and drawbacks of each treatment options, including future perspectives part, in detail.

Response 8. For each treatment option (tissue transplantation, cell transplantation, gene therapy and drug delivery) the therapeutic limitations and drawbacks have been explained in each individual section and, in details, in the future perspectives.

Point 9. Plz, add summarized tables about all therapeutic modalities And also differentiation molecular methods of iPS cells.

Response 9. We agree with the reviewer and two tables with details, follow-up time and clinical outcome of EEC syndrome patients have been added (Table 1 and Table 2).

Point 10. There are articles published in the last year that talk about other therapies that could serve to broaden and update the discussion.

Response 10. As suggested by reviewer, the bibliography has been updated (the added references are highlighted in yellow).

Point 11. Future lines of therapeutic option should be expanded.

Response 11. We believe this point has been already described in the sections reporting future applications with oral mucosal epithelial stem cells, induced pluripotent stem cells, ABCB5+ and mesenchymal stem cells.

Point 12. What is the major difference of EEC and generally known limbal cell deficient diseases, regarding on treatment and prognosis? Why?

Response 12. We thank the reviewer for the interesting question but we believe that this topic is beyond the scope of the present review, since the causes of LSCD are multiple and we focused on the condition of patients affected by EEC syndrome, a rare genetic disease in which LSCD has an inexorable progression, due to the malfunctioning of the p63 gene. In this case, the only possibility to definitely treat the pathology is to correct the genetic defect, as well as for the Aniridia syndrome, where LSCD is caused by mutations in Pax6 gene. In case of LSCD caused, for example, by viral infections, a limbal stem cell transplantation may restore the visual acuity.

Reviewer 2 Report

Cells

Manuscript Number: Cells-2106219

“Innovative Therapeutic Approaches for the Treatment of Ocular Morbidities in Patients with ECC Syndrome”

In the manuscript, “Innovative Therapeutic Approaches for the Treatment of Ocular Morbidities in Patients with ECC Syndrome”. The authors review the current state of the art in the management of the corneal consequences of ECC, a very rare genetic condition. The manuscript is easily comprehensible and well written. It was a very nice read and I think it is an important, though rare and overlooked pathology. I commend the authors for their work in drawing attention to this orphan pathology. Where I have made “suggestions” to change some minor details, this is to assist the flow of the text and many and not mandatory.

Below you will find some point to point comments:

1 – Page 1 line 30: Familiar should be changed to familial

2 – Page 1 line 43: I suggest changing “born dead” to non-viable.

3 – Page 2 line 53: I suggest changing “unproper hydration” to insufficient hydration.

4 – It would be beneficial to add images of the overall ECC phenotype i.e. images of the split hands and cleft palates to aid recognition of these cases in the clinic.

5 – Page 4: A table summarising the data from the case reports and follow up times would be useful for the reader.  

6 – Page 6: I would suggest reducing the data presented from CLET to that of the ECC cases only, the overall success rate is misinformative because when see the ECC cases alone, they are not that good. I think it would be beneficial to highlight the results of the one case from Shortt and three from Daya, separate from the autologous outcomes because they are very different pathologies and outcomes.

7 – Page 8: Line 251 the line “Differently from” Should be changed to “In contrast to N DNp63a, which does not allow any

8 – Page 10: Line 353: oculr should be changed to ocular

End of comments

Author Response

Response to Reviewer 2 Comments

Point 1. Page 1 line 30: Familiar should be changed to familial

Response to point 1. Done.

Point 2. Page 1 line 43: I suggest changing “born dead” to non-viable.

Response to point 2. Done.

Point 3.  Page 2 line 53: I suggest changing “unproper hydration” to insufficient hydration.

Response to point 3. Done.

Point 4. It would be beneficial to add images of the overall ECC phenotype i.e. images of the split hands and cleft palates to aid recognition of these cases in the clinic.

Response to point 4. We agree with the reviewer and we added images of split hands-feet in Figure 2.

Point 5. Page 4: A table summarising the data from the case reports and follow up times would be useful for the reader. 

Response to point 5. We agree with the reviewer and two tables with details, follow-up time and clinical outcome of EEC syndrome patients have been added (Table 1 and Table 2).

Point 6.  Page 6: I would suggest reducing the data presented from CLET to that of the ECC cases only, the overall success rate is misinformative because when see the ECC cases alone, they are not that good. I think it would be beneficial to highlight the results of the one case from Shortt and three from Daya, separate from the autologous outcomes because they are very different pathologies and outcomes.

Response to point 6. We agree with the reviewer and only the results of allo-CLET protocols in patients with EEC syndrome/ectodermal dysplasia have been described in order not to give the readers misinformative outcomes.

Point 7. Page 8: Line 251 the line “Differently from” Should be changed to “In contrast to DNp63a, which does not allow any…”.

Response to point 7. Done.

Point 8.  Page 10: Line 353: oculr should be changed to ocular

Response to point 8. Done.

Round 2

Reviewer 1 Report

This manuscript has been improved and sufficiently addressed in response to the reviewers’ comments.